# Strategies in Transfer Learning for Low-Resource Speech Synthesis: Phone Mapping, Features Input, and Source Language Selection

*Phat Do[1], Matt Coler[1], Jelske Dijkstra[2], Esther Klabbers[3]*

[1]Campus Fryslân, University of Groningen, the Netherlands
[2]Fryske Akademy/Mercator Research Centre, the Netherlands
[3]ReadSpeaker, the Netherlands

{t.p.do, m.coler}@rug.nl, jdijkstra@fryske-akademy.nl, esther.judd@readspeaker.com

## Abstract

We compare using a PHOIBLE-based phone mapping method and using phonological features input in transfer learning for TTS in low-resource languages. We use diverse source languages (English, Finnish, Hindi, Japanese, and Russian) and target languages (Bulgarian, Georgian, Kazakh, Swahili, Urdu, and Uzbek) to test the language-independence of the methods and enhance the findings' applicability. We use Character Error Rates from automatic speech recognition and predicted Mean Opinion Scores for evaluation. Results show that both phone mapping and features input improve the output quality and the latter performs better, but these effects also depend on the specific language combination. We also compare the recently-proposed Angular Similarity of Phone Frequencies (ASPF) with a family tree-based distance measure as a criterion to select source languages in transfer learning. ASPF proves effective if label-based phone input is used, while the language distance does not have expected effects.

**Index Terms**: neural text-to-speech synthesis, low-resource languages, transfer learning, phone mapping, phonological features, source language selection

## 1. Introduction

From the 2010s, research in text-to-speech synthesis (TTS) has shifted towards neural TTS as it produces more intelligible and natural output speech compared to previous paradigms [1]. However, neural TTS requires large amounts of training data, which is hard to come by for low-resource languages (LRLs). One workaround is cross-lingual transfer learning, in which the acoustic model is pre-trained on a language with more ample data (the *"source language"*) before being fine-tuned on the limited data of the LRL (the *"target language"*). This has been studied before by e.g., [2] and [3], but there remain questions about its best practices. Two of such questions are how to best deal with the input mismatch between the languages and how to select the source language that gives the best quality.

Our previous study [4] investigated potential answers to these questions. For the first, we proposed a novel method of phone mapping based on the universal phonological features from the PHOIBLE database [5]. This improved output quality in the study's experiment and thanks to its universality, it can work without requiring linguistic expertise of either the source or target language (except their pronunciation dictionaries). For the second, we proposed a novel criterion: Angular Similarity of Phone Frequencies (ASPF), a measure that compares the similarity between the languages' phone systems. Our experiment results showed that ASPF was more effective than the conventionally-used broad language family classification.

However, these findings came from an experiment with a rather limited setting. For languages, it involved mostly European ones: West Frisian as the target language and Dutch, Finnish, French, Japanese, and Spanish as source languages. Extending to more diverse languages would help validating the applicability of the findings. For the baseline in testing ASPF, it used a binary factor of whether the languages were in the same "broad" language families (e.g., Indo-European, Japonic, and Uralic), following [6]. This can be extended by using a general measure to represent the distance between any two languages across families and branches. [7] explored this idea earlier but did not find sufficient evidence to support it, so we would like to build upon it as a baseline for comparing with ASPF.

In addition, a new method to encode the input to the TTS acoustic model has been studied recently: using vectors of phonological or articulatory features instead of phone labels or graphemes. Originally proposed by [8] to handle zero-shot code-switching in TTS, this method was also useful for cross-lingual transfer learning for LRLs. Since it uses a fixed universal set of features for all languages, it eliminates both the input mismatch problem and the need for phone mapping while also increasing (transfer) learning efficiency. This was thus experimented in [3] for transfer learning in low-resource TTS, but they did not find significant improvements in output quality. However, this could be because they used an autoregressive (based on Tacotron 2 [9]) acoustic model, which may have suffered from its less stable attention training, especially with extremely limited data. Therefore, it would be useful to test this by using a non-autoregressive model, and at the same time extending the scope to more diverse languages.

Accordingly, we aim to make the following contributions:

1) We validate and compare the label-based phone mapping method proposed in [4] and the use of phonological features input in cross-lingual transfer learning for LRLs. We use diverse sets of languages: English, Finnish, Hindi, Japanese, and Russian for source languages, and Bulgarian, Georgian, Kazakh, Swahili, Urdu, and Uzbek for target languages. Section 2 explains the selection of these languages.

2) We validate the idea of using ASPF as proposed in [4] to select the source language in cross-lingual transfer learning and compare it with a general language distance measure.

## 2. Languages and resources used

### 2.1. Selecting target languages and source languages

The phone mapping method and the ASPF measure from [4] are intended to work without requiring linguistic expertise (except pronunciation dictionaries) in the languages involved. Therefore, we wanted to use target languages that we do not have expertise in. Also, we wanted to experiment with actual low-

resource languages (LRLs) rather than simulating low-resource scenarios, in order to ensure the applicability of the findings. Therefore, we used three criteria to choose target languages:

- **Lack of support in TTS:** not supported in the "WaveNet" category of Google's TTS service (in September 2022).
- **Access to automatic evaluation:** supported in the "default" category of Google's Speech-to-Text (Sep 2022), to enable evaluation given the intentional lack of linguistic expertise.
- **Availability of resources:** pronunciation dictionaries were a necessity. There should also be at least roughly 10 minutes of open-access annotated single-speaker training data.

Accordingly, we selected six target languages: Bulgarian (*bg*), Georgian (*ka*), Kazakh (*kk*), Swahili (*sw*), Urdu (*ur*), and Northern Uzbek (*uz*). For source languages, we wanted ones from diverse families, with available pronunciation dictionaries and at least roughly 10 hours of annotated single-speaker data. Accordingly, we selected American English (*en-US*), Finnish (*fi*), Hindi (*hi*), Japanese (*ja*), and Russian (*ru*).

### 2.2. Language resources: dictionaries & data sets

Table 1 details the pronunciation dictionaries and data sets used. All source language data sets have roughly 10 hours of data, while those of target languages have approximately 10 minutes (160-200 utterances). Random sampling was used for data sets that have more data, and we maintained a similar distribution of utterance duration across the source languages. For Common Voice, we only used the "validated" set.

Table 1: *Language resources used*

| Language | Dictionary | Data set |
|---|---|---|
| English (*en-US*) | MFA v2a [10] | LJSpeech [11] |
| Finnish (*fi*) Japanese (*ja*) | ipa-dict [12] | CSS10 [13] |
| Hindi (*hi*) Russian (*ru*) | CV v2 [14] | IndicSpeech [15] M-AILABS [16] |
| Bulgarian (*bg*) Georgian (*ka*) Kazakh (*kk*) Urdu (*ur*) Uzbek (*uz*) | CV v2 [14] | Common Voice 10.0 [17] |
| Swahili (*sw*) | MFA v2a [18] | |

### 3. PHOIBLE-based phone mapping

PHOIBLE [5] is a phonological database of more than 2,000 languages. Each phone is represented by a unique IPA symbol and is connected to a unique set of 37 phonological features associated with its pronunciation (examples in Table 2). Thanks to this, given two phones, we can use their two sets of phonological features to compare their similarity and then use this in the phone mapping method, as proposed in [4].

In cross-lingual transfer learning, there are often phones in the target language that do not exist in the source language. For such phones, the acoustic model cannot take advantage of the source training data and thus has to rely on the limited target data to "learn" their embedding weights. This may seriously limit the output speech quality. Instead of this *"nomap"* scenario, we can map each of these phones to its closest counterpart in the source language. Thus, instead of initializing from scratch, the acoustic model can use the "learned" weights of the mapped phone for fine-tuning. We call this scenario *"map"*.

Following [4], we did the phone mapping as follows: we mapped each target language's phone that needed mapping with the source language's phone having the most similar set of 37 PHOIBLE features. In case of ties, we calculated the frequencies of all phones that immediately preceded and followed the target phone (from all of its occurrences in the target training data). We then did the same for all the tied source phone candidates and calculated angular similarities (Section 5.1) between the target phone and each candidate, for both the front (*ASPF-front*) and back (*ASPF-back*) positions. Then the candidate with the highest averaged similarity (*ASPF-averaged*) was picked to favor more frequently occurring phone sequences.

Table 2 details an example of the phone /o/ in Bulgarian (*bg*). Since there were three candidates in American English (*en-US*) with the same similarity score of 35 (/ɒ/, /ʉ/, and /ʊ/), their ASPF values had to be compared. For an example, the *ASPF-back* value of /ʉ/ (0.326) was calculated between a) the frequency vector of all phones that occur after /ʉ/ in the *en-US* data, and b) that of phones occurring after /o/ in the *bg* data. /ʉ/ was then picked since it had the highest *ASPF-averaged*.

## 4. Phonological Features as Input

Previous studies involving features as input used different feature sets. [8] used a set of 10 multi-valued features: 9 directly from the IPA and 1 accounting for "symbol type", which are converted into 49 binary (one-hot) features. Meanwhile, [3] used 24 binary features derived from the formalism of English sounds in [19] and largely overlap with the convention of PanPhon [20]. Recently, [21] concatenated both the features by PanPhon and those by [8] as this resulted in the closest distance (in the embedding space) between the feature vectors and the embeddings of a well-trained phone-based Tacotron 2 model.

In this work, to facilitate comparisons with the PHOIBLE-based phone mapping, we simply used PHOIBLE's set of 37 phonological features as the feature set. Similar to [8] and [3], we replaced the phone embedding layer of the acoustic model's encoder with a linear layer. This linear layer would then have an input dimensionality of 37 instead of the phone inventory size, and with the output dimensionality unchanged. We call the scenarios of using these features as input *"feature"*.

## 5. Source Language Selection Criteria

### 5.1. Angular Similarity of Phone Frequencies (ASPF)

In our previous work [4], inspired by the use of cosine similarity ($S_C$ or $\cos(\theta)$) to measure similarities between text documents, we used angular similarity (calculable from $\cos(\theta)$) between two languages' vectors of phone frequencies to measure the similarity between their phone systems. We followed this method again in this work. For each language $A$, we extracted its phone set and then its vector of phone frequencies $PF_A$. Then, for the similarity between languages $A$ and $B$, $S_\theta$ between $PF_A$ and $PF_B$ was calculated as follows:

$$S_C(PF_A, PF_B) := \cos_\theta = \frac{PF_A \cdot PF_B}{\|PF_A\|\|PF_B\|}$$

$$S_\theta := 1 - \frac{2 \cdot \arccos(\cos_\theta)}{\pi}$$

This $S_\theta$ is called Angular Similarity of Phone Frequencies (ASPF) and represents the degree of similarity between the two languages from which it was calculated ($0 \leq ASPF \leq 1$).

Table 2: *Mapping Bulgarian's /o/ to one of three candidates in American English: /ɒ/, /ʉ/, and /ʊ/. Different attributes marked in red.*

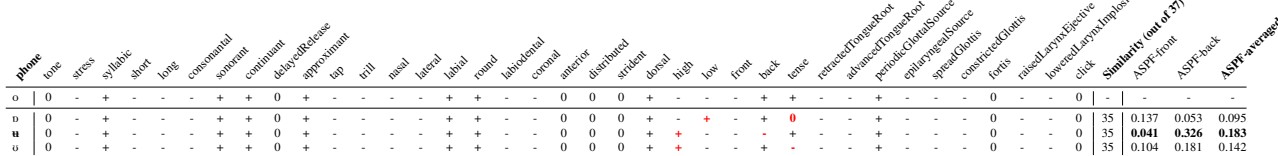

| phone | tone | stress | syllabic | short | long | consonantal | sonorant | continuant | delayedRelease | approximant | tap | trill | nasal | lateral | labial | round | labiodental | coronal | anterior | distributed | strident | dorsal | high | low | front | back | tense | retractedTongueRoot | advancedTongueRoot | periodic-GlottalSource | epilaryngealSource | spreadGlottis | constrictedGlottis | fortis | raisedLarynxEjective | loweredLarynxImplosive | click | Similarity (out of 37) | ASPF-front | ASPF-back | ASPF-averaged |
|---|---|---|---|---|---|---|---|---|---|---|---|---|---|---|---|---|---|---|---|---|---|---|---|---|---|---|---|---|---|---|---|---|---|---|---|---|---|---|---|---|---|
| o | 0 | - | + | - | - | + | + | 0 | + | + | - | - | - | - | + | + | - | - | 0 | 0 | 0 | + | - | - | - | + | - | - | - | - | - | 0 | - | - | - | - | 0 | - | - | - | - |
| ɒ | 0 | - | + | - | - | + | + | 0 | + | + | - | - | - | - | + | + | - | - | 0 | 0 | 0 | + | - | **+** | - | + | **0** | - | - | - | + | - | - | - | - | 0 | - | - | 0 | 35 | 0.137 | 0.053 | 0.095 |
| ʉ | 0 | - | + | - | - | + | + | 0 | + | + | - | - | - | - | + | + | - | - | 0 | 0 | 0 | + | **+** | - | - | **-** | + | - | - | - | + | - | - | - | - | 0 | - | - | 0 | 35 | **0.041** | **0.326** | **0.183** |
| ʊ | 0 | - | + | - | - | + | + | 0 | + | + | - | - | - | - | + | + | - | - | 0 | 0 | 0 | + | **+** | - | - | **-** | - | - | - | - | + | - | - | - | - | 0 | - | - | 0 | 35 | 0.104 | 0.181 | 0.142 |

## 5.2. Distance in language family tree

An earlier work [7] measured similarities between languages by using the "nodes" (families, branches, etc.) in the phylogenetic classification tree from *Ethnologue* [23] as encoded features. This essentially treated the similarity as a categorical variable, which may limit its explanatory power or interpretation in statistical analyses. Recently, [24] more straightforwardly computed the length of the shortest path between the languages, with the unit being the "step" between parent and child. They fruitfully used this to determine nearest languages to aid zero-shot grapheme-to-phone conversion for LRLs. Therefore, we followed this approach and thus the distance between any languages $A$ and $B$ - $dist(A, B)$ - was calculated as:

$$dist(A, B) = D(A) + D(B) - 2 * D(LCA(A, B))$$

where $D(X)$ is the depth of language $X$ (how far it is from the "root", $D(root) = 0$) and $LCA(A, B)$ is the Lowest Common Ancestor of $A$ and $B$. Figure 1 illustrates the family tree used for calculation, which was obtained from Glottolog [22] and only includes the source and target languages we used.

# 6. Experiment details

## 6.1. Data preparation

All utterances were trimmed of leading and trailing silence with a threshold of -35 dFBS. All audio files were mono 16-bit PCM WAV files at 22.5 kHz and conversion was done if needed. We used the Montreal Forced Aligner (MFA) [25] to obtain phone-level alignments between the annotations and audio. All data sets were phonemized using the pronunciation dictionaries in Table 1. These all use IPA symbols so there were no conflicts in phone sets, but they were still manually checked and corrected if needed to ensure consistency. Many of the languages had no available dictionaries that include stress information, so we decided to exclude this from all data. For out-of-vocabulary (OOV) words, we trained and used a grapheme-to-phone (G2P) model for each language with MFA.

## 6.2. Model training

We used the implementation of FastSpeech 2 [26] by [27] for the acoustic models (∼35M parameters), with phone-level pitch and energy prediction, and ground-truth phone duration extracted from MFA like in the original FastSpeech 2 paper. This was used to train all the models in the scenarios of *nomap* and *map*. For *feature*, we modified the encoder as described in Section 4. For waveform synthesis, we used the universal vocoder of HiFi-GAN V1 [28] (∼14M parameters) for all models in the experiments without fine-tuning.

For each source language (*en*, *fi*, *hi*, *ja*, and *ru*), we trained one acoustic model ("*source model*" for short) using phone labels as input, and another one using phonological features as input. Each source model was trained for 300K parameter updates with a batch size of 16 and using the Adam optimizer [29] ($\beta_1 = 0.9$, $\beta_2 = 0.98$, and $\epsilon = 10^{-9}$).

For each target language (*bg*, *ka*, *kk*, *sw*, *ur*, and *uz*), we fine-tuned each of the source models in three different scenarios: *nomap*, *map*, and *feature* (Sections 3 and 4). This resulted in a total of 90 fine-tuned models (6 target languages, from 5 source languages, in 3 scenarios). Each fine-tuning was done for 100K parameter updates with unchanged hyperparameters except for a new batch size of 4. All training was done with one NVIDIA A100 GPU (20GB instance), taking roughly 13.5 hours for each pre-training and 70 minutes for each fine-tuning.

## 6.3. Output evaluation

Intelligibility evaluation was done with Google's Speech-to-Text (STT) service. Each test utterance was run through the STT (using the "default" model, i.e., not "command and search" or "enhanced phone call") to get the automatically transcribed text without converting or post-processing the audio beforehand. The transcribed text was then normalized (removing punctuation marks and converting to lower-case) and used for calculating the Character Error Rate (CER) against the ground-truth text annotation from Common Voice.

For naturalness evaluation using Mean Opinion Score (MOS), research in automatic prediction most notably started with MOSNet [30] and has been building up to the recent Voice-MOS Challenge 2022 [31]. Due to the intentional lack of linguistic expertise in this work, we also used automatic MOS prediction for evaluation. The Challenge had an out-of-domain (OOD) prediction track, where systems were tested on data of a listening test different from the one they were trained on (with some fine-tuning). In this, the baseline system "B01" had strong performance and ranked the fifth and second (out of 18) in terms of Pearson and Spearman correlation, respectively. This was for system-level prediction, which means averaging all predictions per TTS system in order to compare between systems. This scenario lines up well with our use case: it could be considered OOD because we did not have any labeled MOS data and we were mainly interested in system-level comparison. Therefore, we followed this "B01" baseline for our prediction model.

We followed its implementation in [32], which took a large pre-trained self-supervised learning speech model (*wav2vec 2.0 Base* [33]), added a linear layer to the model's output embeddings, and fine-tuned it for MOS prediction using L1 loss on the BVCC data set [34]. Another work of ours [35] experimented further on this and found that fine-tuning such a model further with MOS data from the SOMOS data set [36] led to a statistically significant improvement in performance. We used this model (fine-tuned on BVCC and then SOMOS) to evaluate our TTS models. However, this model still had very limited zero-shot prediction performance at the utterance level, with a median Pearson's *r* correlation of 0.21 (in our independent test set). Its zero-shot system-level performance was, even though not ideal, better with a median *r* of 0.59. Therefore, we only measured and analyzed MOS at the system level in this study.

To verify that the source models had roughly the same baseline quality, we synthesized and evaluated 30 random unseen test utterances from each model. Pilot tests showed that for

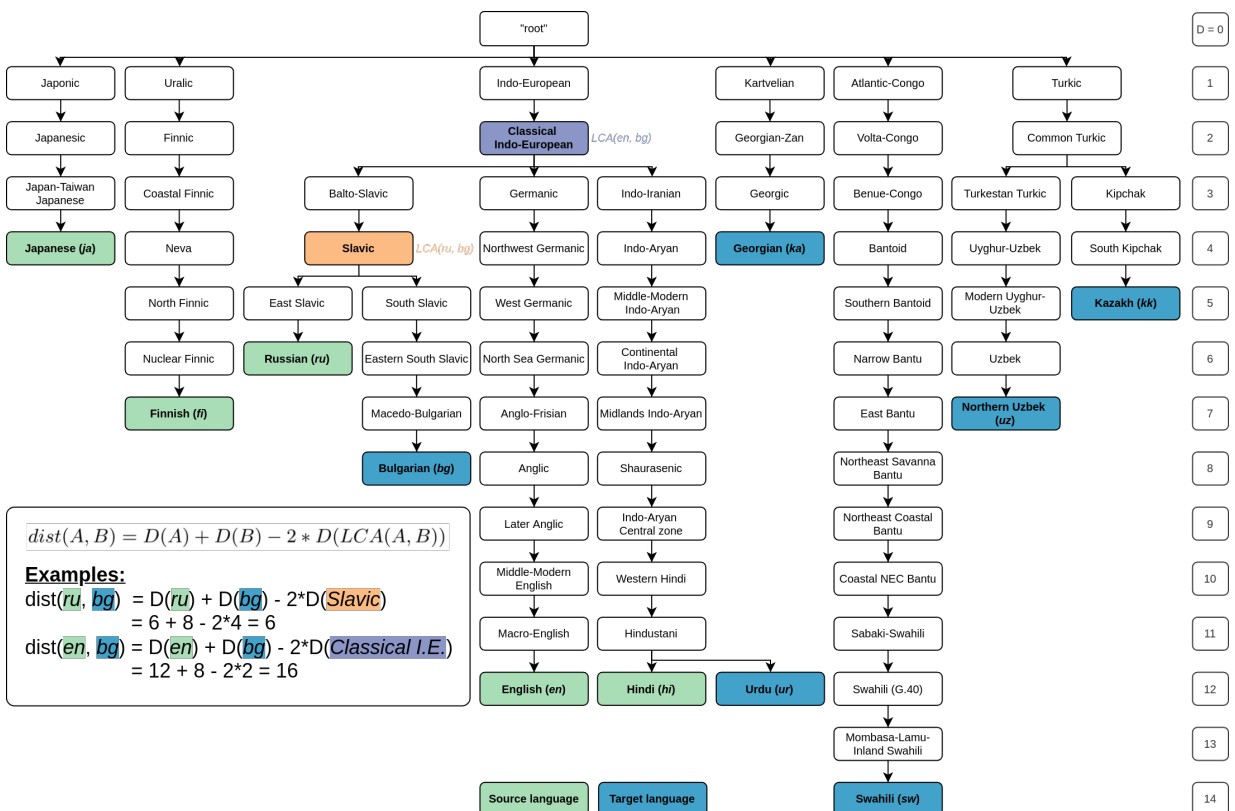

Figure 1: *Language family tree used for calculating distances, extracted from Glottolog [22]*

different languages, Google's STT models understandably had different performance levels, and so did our MOS prediction system. This means there were (unintentional) biases between test languages, so to avoid this, we used an intra-language relative metric for comparison: the difference (in both CER and MOS) between each pair of synthesized and ground-truth utterances. Wilcoxon tests of this metric between the five source languages showed no significant differences.

We wanted to use test sets that were as representative (regarding phone distributions) of the training data as possible. To this end, for each target language, we randomly sampled its available data 10,000 times, each time picking out a set of 100 utterances and calculated their phone frequencies. These were then compared to those of the training data (using the ASPF in Section 5.1), and the set with the highest ASPF was chosen. The resulting test sets all have very high ASPFs, ranging from 0.943 to 0.978. We then synthesized the corresponding 100 test utterances for each of the 90 fine-tuned models described in Section 6.2, and conducted the CER and MOS evaluations. Samples of the synthesized utterances can be found online[1].

## 7. Results & discussion

### 7.1. Effects of phone mapping and features input

The CER (Character Error Rate) data contains repeated measures, as each test utterance had many synthesized versions coming from different fine-tuned models. Therefore, we used mixed effect models [37] to test for the effects of phone mapping and features input on CER, including random intercepts for the test utterances to account for the by-utterance variance. To isolate and highlight the effects being tested, as well as to en-

---

[1]`phat-do.github.io/transfer-SSW23/`

able the comparison in MOS, we separated the analyses into 30 scenarios according to the source and target languages. Table 3 shows the effects of phone mapping and features input.

Compared to *nomap*, *map* significantly decreased CER in 15 scenarios while *feature* did so in 22 out of 30. These effects and the significance codes for their *p*-values are shown in bold. For an example of interpretation, for pre-training on Hindi (*hi*) and fine-tuning on Georgian (*ka*), the mean CER of label-based transfer learning without mapping (*nomap*) was 33.89% and using phone mapping (*map*) decreased it by 3.27 percentage points (p.p.), while using feature-based input (*feature*) decreased it by 9.10 p.p. From the significant effects, the average decrease in CER was 3.48 p.p. for *map* and 4.97 p.p for *feature*.

To confirm that *feature* outperformed *map*, we conducted Wilcoxon rank tests of the CER values between them in groups of target languages, with the alternative hypotheses that the median CERs from *feature* were smaller than those from *map*. Table 4 shows the results, together with the differences in median CERs, confirming that *feature* indeed outperformed *map* for 5 out of the 6 target languages except Urdu ($p = .60$).

For the reasons mentioned in Section 6.3, we only considered the predicted MOS results at the system level: averaging the predictions of all utterances per system and comparing using these mean values. As a result, we could not run statistical tests and thus could only compare the mean values between *nomap*, *map*, and *feature*. As shown in Table 3, compared to *nomap*, both *map* and *feature* improved MOS in most of the scenarios, and *feature* performed the best in 16 out of 30 scenarios.

The analyses of CER and predicted MOS above show that both phone mapping and using features input improved the output speech quality in transfer learning, and the latter were effective in more scenarios and generally outperformed the former. However, the results also showed that they were not always

Table 3: *Effects of phone mapping and features input*
*Signif. codes: 0 '***' 0.001 '**' 0.01 '*' 0.05 '.' 0.1 ' ' 1*

| Tgt | Src | CER (percentage point) | | | | | Predicted MOS | | |
|---|---|---|---|---|---|---|---|---|---|
| | | nomap | map effect | map p | feature effect | feature p | nomap | map | feat. |
| bg | en | 7.79 | 0.27 | | **-1.59** | . | 3.00 | **3.04** | **3.05** |
| | fi | 7.18 | **6.02** | *** | **2.24** | * | 3.00 | 2.95 | 2.95 |
| | hi | 6.70 | 0.44 | | -0.54 | | 3.02 | **3.05** | **3.06** |
| | ja | 11.44 | 1.61 | | **-2.05** | . | 2.96 | **2.97** | 2.96 |
| | ru | 7.34 | **-1.65** | . | **-1.87** | * | 3.00 | **3.00** | **3.07** |
| ka | en | 35.35 | -2.42 | | **-7.47** | *** | 2.51 | **2.57** | **2.61** |
| | fi | 39.49 | **-3.53** | * | -2.22 | | 2.49 | **2.58** | **2.57** |
| | hi | 33.89 | **-3.27** | * | **-9.10** | *** | 2.57 | **2.60** | **2.68** |
| | ja | 43.38 | **-6.75** | *** | **-13.82** | *** | 2.40 | **2.51** | **2.55** |
| | ru | 32.05 | -1.69 | | **-7.82** | *** | 2.43 | **2.55** | **2.60** |
| kk | en | 19.54 | -1.21 | | **-3.93** | * | 2.41 | **2.48** | **2.47** |
| | fi | 23.11 | 0.11 | | **-2.86** | . | 2.38 | **2.39** | **2.43** |
| | hi | 18.83 | **3.16** | . | -2.00 | | 2.37 | **2.39** | **2.45** |
| | ja | 35.72 | **-10.09** | *** | **-9.21** | *** | 2.25 | **2.39** | **2.40** |
| | ru | 21.89 | 0.92 | | **-2.98** | . | 2.33 | **2.38** | **2.40** |
| sw | en | 14.42 | -1.32 | | **-1.75** | . | 2.64 | **2.73** | **2.72** |
| | fi | 18.41 | 0.14 | | -0.43 | | 2.62 | **2.65** | **2.63** |
| | hi | 15.94 | **-1.47** | . | **-3.41** | *** | 2.69 | **2.70** | **2.72** |
| | ja | 21.23 | **-2.30** | . | **-6.26** | *** | 2.58 | **2.64** | **2.62** |
| | ru | 17.59 | **-3.15** | ** | **-4.02** | *** | 2.58 | **2.65** | **2.70** |
| ur | en | 63.48 | 0.87 | | **-3.07** | * | 2.28 | 2.25 | **2.32** |
| | fi | 65.17 | **-6.07** | *** | -0.22 | | 2.24 | **2.25** | 2.23 |
| | hi | 61.92 | **-4.43** | ** | **-3.27** | * | 2.28 | **2.31** | **2.35** |
| | ja | 68.42 | **-7.50** | *** | **-6.54** | *** | 2.26 | **2.30** | **2.30** |
| | ru | 69.14 | **-5.30** | *** | **-7.80** | *** | 2.27 | **2.28** | 2.23 |
| uz | en | 34.55 | 1.11 | | **-5.10** | *** | 2.41 | **2.46** | **2.52** |
| | fi | 39.91 | **-3.01** | . | **-5.14** | *** | 2.38 | **2.45** | **2.42** |
| | hi | 26.77 | -2.09 | | -1.24 | | 2.40 | **2.42** | **2.44** |
| | ja | 40.84 | **-5.37** | *** | **-7.49** | *** | 2.31 | **2.43** | **2.37** |
| | ru | 32.11 | **-4.48** | ** | -1.57 | | 2.37 | **2.41** | **2.39** |

Table 4: *Differences in median CER of "map" and "feature"*

| Target lang. | bg | ka | kk | sw | ur | uz |
|---|---|---|---|---|---|---|
| $M_{map}$ - $M_{ft}$ | **1.75 ** | **6.31 *** | **3.54 ** | **1.11 ** | 0.00 | **1.68 .** |

effective. Extra tests showed that ASPF affected the relative change in CER compared to *nomap*: it decreased this change by 0.78 p.p (*map*) and 0.76 p.p. (*feature*) for every increase of 10 p.p. in ASPF. This means ASPF could explain why *map* and *feature* were not effective for all language combinations.

**7.2. Source language selection criteria: ASPF vs. *dist***

For an overview of the criteria's effects on the whole data, here we used another measure to compare across different target languages: the increase in CER compared to that from the ground-truth audio (*CER_increase_gt*). For example, for a certain test utterance, if the CER obtained from running STT on the ground-truth audio is 2% and that on a synthesized utterance is 5%, the corresponding *CER_increase_gt* will be 3%. We then tested the effects of *ASPF* (Section 5.1) and *dist* (Section 5.2) on *CER_increase_gt* in three different groups: *nomap*, *map*, and *feature*. We used linear mixed effects models with random in-

tercepts for the test utterances and random slopes for the effects being tested. Table 5 shows these effects together with the significance code for their corresponding *p*-values.

Table 5: *Effects of criteria on "CER_increase_gt" (p.p.)*

| Group | ASPF (per 10 p.p.) | | dist (per 1 unit) | |
|---|---|---|---|---|
| *nomap* | **-1.01** | (***) | **-0.48** | (***) |
| *map* | **-0.60** | (**) | **-0.20** | (***) |
| *feature* | -0.11 | | **-0.25** | (***) |

*ASPF* had statistically significant effects on *CER_increase_gt*, decreasing it by 1.01 p.p. and 0.60 p.p. respectively for *nomap* and *map* for every increase of 10 p.p. in ASPF. This confirms its usefulness in selecting source languages, with or without phone mapping: the higher the ASPF (i.e., the more similarity between the target language and the candidate source language), the better the output quality in CER. However, its effect in *feature* was not statistically significant. This may mean that if we use phonological features as input, due to the universality of the feature set and the better transfer learning efficiency, the importance of selecting the "right" source language lessens. However, it may also just mean that ASPF is not effective in this case, and thus should be investigated further in future work.

Although *dist* had statistically significant effects in all groups, their effects were opposite the expectation: the larger the distance (i.e., the less similarity between the languages), the better the output quality in CER. This could mean that even though our distance measure statistically had effects, these effects could have come from another unknown factor that may be somewhat collinear to the distance measure. This should definitely be looked at further in future work, but as of now it remains unsuitable as a criterion to select source languages.

## 8. Conclusions

We validated and compared the PHOIBLE-based phone mapping method proposed in [4] and the use of phonological features input in cross-lingual transfer learning for TTS in low-resource languages (LRLs). We used diverse sets of source languages (English, Finnish, Hindi, Japanese, and Russian) and target languages (Bulgarian, Georgian, Kazakh, Swahili, Urdu, and Uzbek) to enhance the applicability of the findings. We used CER calculated from Google's Speech-to-Text service and MOS from an MOS prediction system for evaluation. Results showed both phone mapping and features input improved the output quality, with the latter performing the best. However, they also depended on the specific language combination.

We also validated the Angular Similarity of Phone Frequencies (ASPF) as proposed in [4] and compared it with a family tree-based distance measure inspired by [24] as a criterion to select source languages in cross-lingual transfer learning. ASPF proved effective in both scenarios of using label-based phone input, while the language distance had effects opposite to expectation. Future research will look further into the latter.

Future work is also planned to compare transfer learning from monolingual source models and from multilingual models, as the latter may benefit from the richer combined phone inventory and thus have better learning efficiency.

**Acknowledgements:** We thank the Center for Information Technology of the University of Groningen for providing access to the Hábrók high performance computing cluster.

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
