# OpenReview forum: "Strategies in Transfer Learning for Low-Resource Speech Synthesis: Phone Mapping, Features Input, and Source Language Selection"
_Interspeech.org/2023/Workshop/SSW — SSW12_

### Official Review · Reviewer_aLyR · 2023-06-05
**A nice comparison through a well-designed experiment**

**Rating:** 8
**Confidence:** 4

**Review:**

This paper investigates transfer learning strategies for the low-resource TTS development. PHOIBLE is used as a universal phoneme representation. Then, several approaches to address a phone mismatch issue between different languages have been investigated, such as no mapping, a phoneme mapping based on ASPF, and the direct use of PHOIBLE's distributed representation. In addition, source language selection approaches have also been investigated, such as the use of ASPF or the distance on the language family tree as a selection measure. The experimental conditions were carefully designed so as to cover wide varieties of the source and target languages. The experimental results are supposed to be reasonable and informative for our research community. This paper is well written. It will be worthwhile to present this work in the workshop.

In Table 3, the result of transfer learning from Finnish to Bulgarian has a different tendency from those of the others, i.e., both map and feature caused performance degradation from nomap. It is nice to deeply investigate the result of this language pair. It would be great if the authors could find what caused this degradation. Moreover, it would be nice to more analyze the results of Table 3. If these performance improvements could be explained with some factors, it would be also useful to consider them in the transfer learning.

As mentioned in the conclusions, it is worthwhile to extend this work to a multilingual framework. It is also interesting to think about how to incorporate a specific knowledge, such as the language family tree, for the network training process. It would be nice if it could be effectively used as prior information or regularization in the transfer learning.

---

### Official Review · Reviewer_DZNU · 2023-06-09
**Our of my field, not reviewed.**

**Rating:** 7
**Confidence:** 1

**Review:**

This work is out of my field of expertise. Not reviewed. Please discard the rating provided.

---

### Decision · Program_Chairs · 2023-06-14

**Decision:**

Accept

**Comment:**

SSW2003 received 45 papers. The acceptance rate is 82%. We are pleased to inform you that your paper has been accepted by the SSW2023 Program Committee. Please read the reviews carefully and submit your camera-ready paper by June 28th. Most reviewers performed a detailed review. Please answer to their questions and consider their comments. Note that camera-ready papers are credited with one extra page to allow authors to consider reviewers’ suggestions. So max 7 pages in total including figures & refs.
The deadline for submitting the revised version (with full non-anonymized authors and refs!) is 28th June.